# Identification of miRNA Master Regulators in Breast Cancer

**DOI:** 10.3390/cells9071610

**Published:** 2020-07-03

**Authors:** Antonio Daniel Martinez-Gutierrez, David Cantú de León, Oliver Millan-Catalan, Jossimar Coronel-Hernandez, Alma D. Campos-Parra, Fany Porras-Reyes, Angelica Exayana-Alderete, César López-Camarillo, Nadia J Jacobo-Herrera, Rosalio Ramos-Payan, Carlos Pérez-Plasencia

**Affiliations:** 1Laboratorio de Genómica, Instituto Nacional de Cancerología, Tlalpan, CDMX 14080, Mexico; maga94@comunidad.unam.mx (A.D.M.-G.); dfcantu@gmail.com (D.C.d.L.); oliver.millan.sg@gmail.com (O.M.-C.); jossithunders@gmail.com (J.C.-H.); adcamposparra@gmail.com (A.D.C.-P.); 2Servicio de Anatomía Patológica, Instituto Nacional de Cancerología, Tlalpan, CDMX 14080, Mexico; car_plas@yahoo.com; 3Decanato, Ciencias de la Salud, Universidad Autónoma de Guadalajara, Jal Zapopan 45129, Mexico; drako098@hotmail.com; 4Posgrado en Ciencias Biomédicas, Universidad Autónoma de la Ciudad de México, CDMX 03100, Mexico; genomicas@yahoo.com.mx; 5INCMNSZ, Unidad de Bioquímica, Tlalpan, CDMX 14080, Mexico; nadia.jacobo@gmail.com; 6Faculty of Biology, Autonomous University of Sinaloa, Culiacán 80007. Sin, Mexico; rosaliorp@uas.edu.mx; 7Laboratorio de Genómica, Unidad de Biomedicina, FES-IZTACALA, UNAM, Tlalnepantla 54090, Mexico

**Keywords:** miRNA, breast cancer, master regulator, integrative analysis, TCGA data analysis

## Abstract

Breast cancer is the neoplasm with the highest number of deaths in women. Although the molecular mechanisms associated with the development of this tumor have been widely described, metastatic disease has a high mortality rate. In recent years, several studies show that microRNAs or miRNAs regulate complex processes in different biological systems including cancer. In the present work, we describe a group of 61 miRNAs consistently over-expressed in breast cancer (BC) samples that regulate the breast cancer transcriptome. By means of data mining from TCGA, miRNA and mRNA sequencing data corresponding to 1091 BC patients and 110 normal adjacent tissues were downloaded and a miRNA–mRNA network was inferred. Calculations of their oncogenic activity demonstrated that they were involved in the regulation of classical cancer pathways such as cell cycle, PI3K–AKT, DNA repair, and k-Ras signaling. Using univariate and multivariate analysis, we found that five of these miRNAs could be used as biomarkers for the prognosis of overall survival. Furthermore, we confirmed the over-expression of two of them in 56 locally advanced BC samples obtained from the histopathological archive of the National Cancer Institute of Mexico, showing concordance with our previous bioinformatic analysis.

## 1. Introduction

Breast cancer (BC) is the leading type of neoplasia in the female population, with two million new cases diagnosed worldwide each year [1]. Advanced clinical stages are related to higher mortality rates due to their metastatic behavior [2]; therefore, it is necessary to investigate and understand the molecular mechanisms underlying the metastatic cancer phenotype.

Tumor progression is the tendency of tumoral cells to acquire traits—usually known as hallmarks—such as the loss of cellular differentiation, an increased cell cycle, invasion, and metastases that lead to a more aggressive phenotype [3]. One class of molecules known to be involved in tumor progression are the microRNAs or miRNAs. These molecules are small non-coding RNAs of 21 to 23 nucleotides, whose main function is the post-transcriptional regulation of a whole range of genes and are key regulators of oncogenic pathways [4]. In BC, their pathogenic participation has been extensively investigated, showing that these molecules display an aberrant expression pattern that can distinguish between normal and BC tumor tissues [5]; they can also successfully identify different subtypes of BC [6], and are involved in the transition of specific key events during the progression of the disease [7]. Likewise, they have been successfully used as prognostic biomarkers in invasive ductal carcinoma tumors [8] and in specific subtypes such as the triple negative [9]. Due to their multitarget nature miRNAs can maintain the major characteristics of cancer phenotype; such kind of miRNAs are known as master regulators (MMR) [10].

The use of high-throughput data and the inference of miRNA–mRNA networks is a bioinformatic tool that allows us to understand the complex participation of miRNAs in cancer (reviewed in [11]). For instance, in serous ovarian cancer, it was found that eight miRNAs were able to regulate 89% of genes associated with the mesenchymal phenotype [8]. Meanwhile, in BC, it was found that miRNAs act as co-transcriptional modules that could be both positively and negatively associated to their target mRNAs [12]. Another study by Cantini et al., showed that the miR-199/miR-214 cluster of polycistronic miRNAs was acting as an MMR by regulating the epithelial to mesenchymal transition [13]. Another example of an integrated analysis of miRNAs and expression profiles reported a set of miRNA master regulators in colorectal cancer, which were responsible for the regulation of this cancer subtype [10]. In the present work, our major aim was the identification of miRNAs acting as master regulators that were consistently upregulated along the clinical stages of BC. To achieve this, we downloaded from TCGA miRNA and mRNA sequencing data corresponding to 1091 BC patients and 110 normal adjacent tissues. The expression data were compared between tumors and normal tissues. We obtained 209 consistently over-expressed miRNAs along the different cancer stages. Of them, 61 miRNA were considered master regulators in the BC transcriptome through inferring their miRNA–mRNA network using the ARACNe algorithm [14]. Next, we calculated the oncogenic activity of each (61) miRNA to the breast tumor by considering the number and expression profile of validated oncogenes and tumor suppressors genes they regulate. This group of miRNAs was involved in the regulation of pathways such as cell cycle, PI3K–AKT, DNA repair, k-ras signaling, among others. Using univariate and multivariate analysis, we found that five of these miRNAs could be used as biomarkers related to the overall survival rate. Furthermore, we validated the overexpression of three of them in locally advanced BC samples obtained from the histopathological archives of the National Cancer Institute of Mexico, showing concordance with our previous bioinformatic analysis.

## 2. Materials and Methods

### 2.1. Analysis of Upregulated miRNAs in Clinical Stages

The general workflow is shown in Figure 1. To identify the consistently upregulated miRNAs in breast cancer, we first obtained the hiseq read counts from the TCGA BC project [15] corresponding to 1091 BC patients (Stages I–IV) and 110 normal adjacent tissues using the Bioconductor package TCGABIOLINKS [16]. Differential expression analysis was carried out using DESeq2 [17]. Only miRNAs with an FDR ≤ 0.05 and a log2FoldChange of at least +0.5 were considered. The correlation between miRNA expression and TNM stages defined by the American Joint Committee on Cancer (AJCC) was done using Spearman correlation, included in the R package Hmisc. We only considered miRNAs with a *p*-value of ≤ 0.05 and a positive correlation coefficient.

### 2.2. Identification of miRNA Master Regulators

To determine the number of miRNAs acting as master regulators in BC, we obtained mRNAs differentially expressed in tumor tissues employing TCGABIOLINKS and DESeq2. Only mRNAs with an FDR ≤ 0.05 were considered for this analysis. Subsequently, mRNA and miRNA datasets were normalized using the VST method implemented in the DESeq2 package. Next, we reconstructed the miRNA–mRNA network with ARACNe-AP [14] employing our previously filtered data consisting of the overexpressed miRNAs and the differentially expressed mRNAs, using 100 bootstrap replications. The miRNA–mRNA interactions in the resulting network were filtered based on their presence in at least one predicted (DIANA-microT-CDS, ElMMo, MicroCosm, miRanda, miRDB, PicTar, PITA and TargetScan) and/or experimentally validated databases (miRecords, miRTarBase, TarBase) using the Bioconductor package multiMir [18]. With this approach, we ensured that the resulting network contained both inferred and experimentally validated data. Next, we employed the master regulator inference algorithm [19] included in the Bioconductor package [20]. miRNAs were assigned as MMR if they had an FDR ≤ 0.05. Finally, the network analysis was carried out with Cytoscape v3.7.1 (National Institute of General Medical Sciences (NIGMS), Bethesda, MD, USA) [21].

### 2.3. Oncogenic Activity of miRNAs MMR

Any miRNA has the ability to regulate both oncogenes and tumor suppressor genes. Hence its oncogenic activity (OA) is the likelihood that a given miRNA will regulate an oncogene and a tumor suppressor gene in both ways i.e., upregulate an Oncogene, down-regulate a tumor suppressor gene or vice versa. To obtain the oncogenic activity for each miRNA, we first downloaded the manually curated oncogene/tumor suppressor data from the Catalogue of somatic mutations in cancer (COSMIC [22]).

Therefore, we first calculated its Oncogenic Effect (OE) as the sum of upregulated oncogenes and down-regulated TSG, which is represented with the following formula:(1)$$O_{e}=\left(U_{ONC}+D_{TSG}\right)-ONC_{TSG}$$
where: $\;O_{e}=$ Oncogenic effect; $U_{ONC}=$ Upregulated oncogenes; $D_{TSG}=$ Downregulated tumor suppressor genes; $ONC_{TSG}=$ Genes with dual function as oncogenes and tumor suppressors.

Then, the Anti-oncogenic Effect (AE) as the sum of up-regulated TSG and down-regulated oncogenes:(2)$$A_{e}=\left(D_{ONC}+U_{TSG}\right)-ONC_{TSG}$$
where:$\;A_{e}=$ Antioncogenic effect; $D_{ONC}=$ Downregulated oncogenes; $U_{TSG}=$ Upregulated tumor suppressor genes; $ONC_{TSG}=$ Genes with dual function as oncogenes and tumor suppresors.

Finally, the OA is the relation between AO-OE and the total number of possible driver targets for each miRNA:(3)$$O_{A}=\frac{O_{e}-A_{e}}{T}$$
where:$\;O_{A}=$ Oncogenic activity; $O_{e}=$ Oncogenic effect; $A_{e}=$ Antioncogenic effect; T= Total number of targets (oncogenes, tumor suppressor genes and genes with dual function).

### 2.4. Pathway Analysis

The pathway analysis was done using the WebGestalt platform [23] (http://www.webgestalt.org); we uploaded the differentially expressed mRNA gene symbols and their log2FoldChange. Statistically significant pathways were considered with an FDR ≤ 0.05.

### 2.5. Intrinsic Subtype Classification and Differential Expression

The PAM50 subtype classification of the TCGA samples was assessed using the Bioconductor package Genefu [24]. Differential expression between subtypes was carried out using the Bioconductor package DESeq2 [17], where miRNAs with an FDR ≤ 0.05 were considered as significant.

### 2.6. Survival Analysis

Clinical data was obtained from the 1091 BC patients analyzed from the TCGA. Next, we divided the patients into two groups, high and low expression based on the median expression of each MMRs. Kaplan–Meier curves and Cox regressions were obtained in the R environment using the survival package, the multivariate Cox regressions were adjusted using the age and clinical stages of the patients. Statistical significance was assessed using log-rank test < 0.01.

### 2.7. Patient Samples

A prospective group of 56 patients diagnosed with locally advanced BC were enrolled. Formalin-fixed paraffin-embedded (FFPE) tissue blocks were obtained from the Pathology department at the Instituto Nacional de Cancerología. All patients included in the present study signed an informed consent form; protocol number (015/03/019/ICI).

### 2.8. RNA Isolation and Real-Time qPCR

Total RNA was extracted from the FFPE tissue blocks using the miRNeasy FFPE Kit (Cat. No. 217504 Qiagen. Hilden, Germany). RNA was quantified in Qubit 3.0 Fluorometer with Qubit RNA HS Assay Kit (Cat. No. Q32852. Thermo Fisher Scientific. Waltham, MA, USA). Reverse transcription was performed using 10 ng of RNA with TaqMan Advanced miRNA cDNA Synthesis Kit (Cat. No. A28007. Thermo Fisher Scientific) following the manufacturer’s recommendations. Quantitative real time PCR was performed with TaqMan Advanced miRNA Assays for: hsa-miR-940, hsa-miR-1307-3p, hsa-miR-340-3p using hsa-miR-16 as an endogenous control (Cat. No. 4331182, Thermo Fisher Scientific). q-PCR reaction was performed at 95 °C for 20 s and at 95 °C for 1 s (40 cycles) with Taqman Fast Advanced Master Mix (Cat. No. 4444557, Thermo Fisher Scientific) on Step One System (Thermo Fisher Scientific). All reactions were performed in triplicate and fold changes were calculated using the 2–ΔΔCT method. The Mann–Whitney U test was used to identify significant miRNAs, considering *p* < 0.05 as statistically significant.

### 2.9. Tissue Expression Meta-Analysis

Antibody-based proteomic data of 2 normal breast tissues and 11 BC samples were obtained from The Human Protein atlas [25], where the staining activity was compared between the two groups.

## 3. Results

### 3.1. Identification of miRNAs Consistently Upregulated in Breast Tumor Tissues

Our aim was to get a list of miRNAs consistently upregulated in BC (Stages I–IV) obtained from the TCGA breast cancer project. We selected only those miRNAs that had an FDR < 0.05. In total, 217 miRNAs constantly upregulated in all clinical stages (Figure 2) were found. To verify the upregulation of these miRNAs, expression levels were correlated to the clinical stages using the Spearman correlation. From the last analysis, 204 miRNAs had a significantly positive correlation (Appendix A).

### 3.2. Identification of MMRs

Employing the 204 significantly upregulated BC miRNAs, we analyzed their role in the development of the tumor phenotype, so we proceeded to investigate their possible targets and pathways. To do that, we first inferred the tumor miRNA–gene network with an information theory approach, using the mutual information algorithm ARACNe [14] and the differentially expressed genes. Given previous reports demonstrating the ability of miRNAs to upregulate their targets [26,27], we considered both downregulated and upregulated genes in tumoral tissues. The resulting regulatory network consisted of 676,784 miRNA–gene interactions (Appendix A). Next, we filtered the network based on the presence of the miRNA–gene interaction inferred by prediction algorithms. We considered interactions present in at least one database (DIANA-microT-CDS, ElMMo, MicroCosm, miRanda, miRDB, PicTar, PITA and TargetScan) and/or experimentally validated databases (miRecords, miRTarBase, TarBase), since it has been observed that the union of different prediction algorithms have a higher sensitivity and precision than the intersection [28]. This step permitted us to reduce the network to 36,205 miRNA–gene interactions (Appendix A).

Finally, we used the filtered network to infer the MMR using the master regulatory inference algorithm (MARINa) [19] included in the VIPER R package [20]; only those miRNAs with an adj-p value < 0.05 were considered. The results revealed that the tumoral transcriptome could be regulated by 61 putative MMRs (Figure 3), involved in 16,444 interactions, consisting of 6530 (39%) out of 14,449 total genes found to be differentially expressed in the transcriptome of breast tumors (Appendix A). Although each miRNA had both up- and downregulated targets, the main activity and normalized enrichment score (NES) of all MMRs was the downregulation pattern of their targets, as shown in Figure 3 (Appendix A).

Subsequently, we were interested in identifying which targets and miRNAs had the largest number of interactions, so we topologically analyzed the network to further obtain the node degree, a metric that reflects the number of links per node involved in the network. We observed that miR-106b-5p was a MMR with the highest number of total targets present in tumors, followed by miR-590-3p, miR-93-5p, miR-671-5p, and miR-939-5p (Table 1). On the other hand, the genes with the higher number of miRNAs regulating them were RUNX1T1, BNC2, TNS1, DLC1, and FOXP2. As shown in Table 1, of the 20 most regulated genes, only RACGAP1 had a positive fold change in tumors, while the remaining genes showed a downregulation pattern.

### 3.3. Pathway Analysis

For the sake of knowing the biological processes that could be altered by the MMRs, we did a pathway analysis using two of the most-used pathway databases, KEGG and Wikipathways cancer database. Therefore, we uploaded the 6530 unique genes regulated by the MMRs. In Figure 4A,B, we show the activity of the MMRs predicted to be involved in the regulation of multiple oncogenic associated pathways such as cell cycle, Ras signaling, PI3K–AKT–mTOR, DNA damage response, and p53 pathway. A concordance between both gene annotation databases was found in three pathways, cell cycle, Ras signaling, and pathways involved in DNA damage response. It should be noted that of the 6530 unique genes, only 2542 and 862 genes were annotated in the KEGG and Wikipathway databases, respectively. Such numbers represent 38.9% and 13% of the total uploaded genes; thus, the biological processes in which the remaining genes that are not currently annotated could be participating in remain unknown.

### 3.4. Prediction of miRNAs Oncogenic Activity

We reasoned that each MMRA oncogenic contribution to the tumor cell could be measured as the balance of validated driver oncogenes and tumor suppressor genes they could regulate. We termed this balance as the oncogenic activity, which was calculated for each miRNA as mentioned in the Methods section. It should be noted that per se the mutual information measure between miRNA and mRNA does not give the directionality of the regulation (positive or negative); thus, we used the log2FoldChange as a measure of directionality between each miRNA and their targets. The COSMIC database contained 723 manually curated oncogenes and tumor suppressor genes, of which 336 of them were present in our inferred miRNA–mRNA network.

As shown in Figure 5, we found that, of the 61 MMR, 14 had a positive oncogenic activity, 35 had a negative oncogenic activity, and 12 of them had a neutral oncogenic activity. We noticed that the miRNAs with the highest oncogenic activity were miR-3200-3p, miR-185-3p, miR-3187-3p, and miR-210-3p, while the miRNAs with the lowest net oncogenic activity were miR-4677-3p, miR-4326, miR-769-5p, miR-103a-2-5p, and miR-3677-5p. This result suggests that each MMR has negative or positive participation in the tumoral phenotype, whereas a high proportion of them act predominantly as tumor suppressor miRNAs.

### 3.5. Participation of miRNAs Acting as Oncogenic Drivers

Next, we wanted to know the tumor mechanisms and the driver genes regulated by the 14 MMMRs that had a positive contribution to the tumoral phenotype. As shown in Figure 6A, the 14 MMRS regulate 144 gene drivers. Moreover, the top four most highly connected MMRs were commonly involved in invasion and metastasis and resisting cell death, although there were specific hallmarks regulated by each OncoMMRs (Figure 6B).

The top four most highly connected OncoMMR were miR-106b-5p, followed by miR-590-3p, miR-671-5p, and miR-106a-5p with 51, 44, 37, and 18 genes, respectively (Table 2). We observed that OncoMMRs with the higher oncogenic activity had the smallest number of regulated targets.

With respect to the top tumor driver genes that were more regulated by OncoMMRs, these were the tumor suppressor genes FAT homolog 4 (FAT4), Kruppel like factor 6 (KLF6), Rho guanine nucleotide exchange factor 10 (ARHGEF10), and the RB transcriptional corepressor 1 (RB1); all of them were downregulated in tumoral tissues. Moreover, the oncogenes High mobility group AT-Hook 1 (HMGA1), Forkhead box A1 (FOXA1), and the Serine and arginine rich splicing factor 2 (SRSF2) were (as expected) over-expressed (Appendix A).

Interestingly, there was a group of genes co-regulated by all four OncoMMRS. Of these, the tumor suppressor ARHGEF10 [29] and the oncogene SRSF2 [30] were commonly regulated by three (miR-106b-5p, miR-590-3p, and miR-671-5p) and four (miR-106b-5p, miR-590-3p, miR-671-5p, and miR-106a-5p) OncoMMRs, respectively, in the network (Appendix A). As shown in Figure 7A, all three OncoMMR exhibited a negative correlation with the tumor suppressor ARHGEF10. Moreover, all of the four OncoMMRs showed a positive correlation with the oncogene SRSF2, where the largest correlation coefficient was observed for miR-106b-5p with 0.419. Moreover, both ARHGEF10 and SRSF2 showed the same trend of expression at the protein level in breast cancer tissues, as shown in Figure 7C,D.

### 3.6. Identification of miRNA Master Regulators with Clinical Relevance

As all MMR were upregulated in all clinical stages, we reasoned that they could be used as survival biomarkers in breast cancer patients. To achieve this, TCGA patients were grouped in high and low expression for each MMR. Kaplan-Meier curves with log-rank p showed that 5 MMR (miR-151a-5p, miR-340-3p, miR-877-5p, miR-940, and miR-1307-3p) had significant differences between the two groups. The results showed that for all five MMR, patients with a higher expression had a worse overall survival (Figure 8), although the impact of the MMRs in the survival varied depending of the intrinsic subtypes analyzed, where miR-151a-5p showed significative differences in the Normal-like subtype (*p* = 0.0024), miR-877-5p in the Luminal A subtype (*p* = 0.043), miR-940 in the Normal-like subtype (*p* = 0.0014) and miR-1307-3p in the Luminal A subtype (*p* = 0.048) and the Normal like subtype (*p* = 0.0001), whereas miR-340-3p showed no significative differences (Appendix A).

Univariate and Multivariate cox regressions using age, clinical stages, estrogen receptors (ER), and progesterone receptors (PR) status as covariates further suggested that miR-1307-3p could be used as prognostic factor of the overall survival for patients with breast cancer (Table 3).

### 3.7. Impact of MMRs in the BC Intrinsic Subtypes

Next we evaluated the expression of the five MMRs in the intrinsic subtypes of BC based in the PAM-50 classification, as we observed differences in the survival of patients depending on the subtype and considering that the intrinsic subtypes show different molecular profiles that impact on clinical characteristics such as the patient outcome [31]. This analysis showed two main results: first, all of the five MMRs are overexpressed in the subtypes when compared with normal adjacent tissue (Figure 9A–E), which is in accordance with our main hypothesis. Second, there are differences in the expression of the MMRs between subtypes. In the case of miR-151a-5p, we observed no significant differences between the Her2 vs. normal-like and Her2 vs. Luminal B subtypes, where the expression of this MMR is lower and significant in the basal subtype when compared with the other subtypes (Figure 9A).

For miR-340-3p, there were no significant differences in the basal vs. Her2 and Luminal B vs. normal-like subtype; moreover, this miRNA was overexpressed in the Luminal A vs. normal-like and Luminal A vs. Luminal B comparisons (Figure 9B). This pattern was also observed for miR-877-5p, miR-940 and miR-1307, where the comparisons between Luminal A vs. normal-like, Luminal A vs. Luminal B showed an overexpression of the miRNAs, whereas Luminal B vs. normal-like was not significant (Figure 9C–E).

We observed that miR-940 had the largest number of non-significant differences between the subtypes, thus making the expression homogeneous between certain subtypes (Figure 9D). On the other hand, miR-877-5p had the largest number of significant differences, which shows a heterogeneous expression between the subtypes (Figure 9C). Interestingly, the results show that aside from the normal tissue, all five MMRs had a higher expression exclusively in the Luminal A when compared to the other subtypes (Figure 9A–E and Appendix A).

### 3.8. Validation in Locally Advanced Tumoral Tissues

For this study, we wanted to validate the expression profile of three of the univariate and multivariate significant MMRs in an independent cohort. Therefore, we verified the expression level of miR-340-3p, miR-940, and miR-1307-3p in the paraffin-embedded breast tumoral tissue of locally advanced samples. Samples were collected from 56 patients from the National Cancer Institute of Mexico, composed of 8 Her2, 20 Luminal A, 14 Luminal B, and 13 triple negative. As shown in Figure 10, in the comparison between normal vs. tumoral tissue, miR-940, and miR-1307-3p were significantly overexpressed in tumoral tissues, showing the same trend as our previous bioinformatic analysis. Next, we were interested if the trend of MMR overexpression was constant in the molecular subtypes. In the case of miR-1307-3p, we observed no significant differences between the molecular subtypes, meaning that it was constantly overexpressed (Figure 10A), Moreover, we observed that miR-940 had a higher expression in the triple-negative subtype compared to the Luminal A and B (Figure 10B), whereas miR-340-3p showed no significant differences in tumors or in the intrinsic subtypes (Figure 10C). This result makes it feasible to use miR-1307-3p and miR-940 as biomarkers, independent of the molecular subtype.

## 4. Discussion

The system’s biological perspective of cancer is the result of increased entropy, e.g., an increase in cell disorder, leading to aberrations at the genetic and epigenetic level affecting the transcriptional and proteomic profiles of tumor cells [32]. In this chaotic context, the cancer cells must maintain their cellular functions despite the multiple aberrations and insults they are subjected to, preserving the cell robustness by multiple mechanisms such as the post-transcriptional regulation exerted by miRNAs [33]. Previous evidence has demonstrated that, although tumors could have different somatic mutational profiles, they present similar dysregulated transcriptomes [34], implying the presence of stabilizers that allow the maintenance of the cellular functions reliant on the transcriptome. It has been proposed that these master regulators are miRNAs, molecules capable of maintaining the cell transcriptome independent of the events that led to tumor development and the aberrations that the tumor cell is constantly subjected to. All of this confers robustness to biological processes by reinforcing transcriptional profiles and stabilizing expression levels of aberrant transcripts, mainly in copy numbers [35].

Previous studies have hypothesized the presence of miRNAs acting as master regulators in cancer [10], with further studies reinforcing this hypothesis by showing their involvement in co-transcriptional modules in breast tumors [12]. Later, several theoretical and experimental studies confirmed a hypothesis in ovarian [36], colorectal [37], and breast cancers [13]. However, the main pitfalls for studies involving similar computational pipelines are first that they disregard biological information present miRNA–mRNA validated databases. Secondly, they do not take into account the interaction with validated driver oncogenes/tumor suppressors due to its prominent role in carcinogenesis. To our knowledge, this work represents the first effort to elucidate a set of MMRs constantly upregulated in every clinical stage in breast cancer and its relation to the oncogenes/tumor suppressors they regulate.

In this research, we aimed to find the MMRs consistently upregulated in the carcinogenesis of breast tumors and correlate their participation to the oncogenic transformation. To do so, we integrated miRNAs and mRNA data from the TCGA breast project through adaptation of the Master Regulator Inference algorithm (MARINA), an algorithm that was formerly approved inferring aberrant proteins [20] and transcriptional master regulators. [38].

We observed 61 MMRs present consistently in all stages of breast cancer, which were associated with the regulation of 39% of the differentially expressed genes in the breast cancer transcriptome. As expected, the major pathways regulated by the MMR panel are hallmarks of cancer such as the cell cycle and PI3K–AKT–mTOR pathway and DNA repair, among others, as previously reported in several studies (Figure 4). Although BC is a heterogeneous disease, this result reinforces the concept and participation of MMRs that could be regulating a common “core” of genes shared by every intrinsic subtype of BC and whose main function is maintaining the core mechanisms and pathways that a tumor needs in every step of the tumorigenesis and that differentiates them from normal healthy tissue, such as the hallmarks of resisting cell death, sustaining proliferative signaling, enabling replicative immortality and evading growth suppressors.

One of the current challenges when analyzing miRNAs is their classification as oncomiRNAs or tumor suppressor miRNAs due to their diverse nature [39]. In this work, we addressed this problem by calculating the individual miRNA oncogenic activity, a measure proposed by Svoronos et al. [39]. To solve it, we exclusively took those identified driver genes proved to be causal of the development of cancer. When we focused on the regulation of just this class of genes our results showed that most of the miRNAs were acting as a tumor suppressors, which is consistent with other publications (reviewed in [40]). This outcome could suggest that although the tumoral cell has an uncontrolled proliferation tendency, there are fine-regulating mechanisms—including miRNAs—that are involved in the balance of proliferation (Figure 5). However, proving this in vivo and in vitro validation is still necessary.

Our findings revealed a group of MMRs that were acting predominantly as OncoMMRs involved in the maintenance of tumoral phenotype. In this analysis, we focused on the top four OncoMMRs that had the largest number of targets (miR-106b-5p, miR-106a-5p, miR-671-5p, and miR-590-3p) (Figure 6). The expression levels of this OncoMMRs have been widely associated with increased cell proliferation, migration, invasion, and radioresistance [41,42,43,44,45,46] showing concordance with our results. Moreover, the top-four OncoMMRs mentioned before were highly enriched in genes involved in invasion and metastasis, suggesting a co-participation in this hallmark.

Regarding genes regulated by MMRs, we discovered a redundant regulation of the tumor suppressors FAT4 and KLF6, among others, whose loss of expression has been associated with a more aggressive phenotype in breast cancer (Table 2) [47,48,49]. Moreover, we found that the tumor suppressor ARHGEF10 and the oncogene SRFS1 were negatively and positively co-regulated by miR-106b-5p, miR-106a-5p, miR-671-5p, and miR-590-3p, whereas this pattern was observed at the protein level in breast cancer tissues (Figure 7), validating our results. Even though these results were obtained with a limited number of samples, it deserves future analysis to corroborate the correlation between the expression of ARHGEF10 and SRSF2 in a large number of tumor tissues. Recent reports suggest that ARHGEF10 participates as a tumor suppressor involved in the activation of apoptosis during DNA damage [29]. Thus, our outcomes propose that this TSG is subjected to co-regulation by the Onco-MMMRs miR-106b-5p, miR-671-5p, and miR-590-3p. On the other hand, the splicing factor SRSF1 has been observed to be upregulated in breast tumors acting as an oncogene by producing impaired splicing isoforms of the pro-apoptotic genes BCL2 Like 11 (BIM) and the Bridging integrator 1 (BIN1) which are involved in the tumoral phenotype. However, it should be noted that these possible interactions between MMRs and driver genes need to be further confirmed by in vitro experiments.

miRNAs have been proposed as molecular biomarkers in numerous investigations [7,8,9,50], due to their specific expression, stability, and suitability for measurements in peripheral blood. Therefore, we searched for MMRs associated with patient prognosis. We observed that patients with a higher expression of miR-1307-3p, miR-940, and miR-340-3p had a worse overall survival, suggesting their potential use as biomarkers when evaluated on all tumors (Figure 8), whereas miR-1307-3p and miR-940 also showed an over-expression in an independent breast cancer cohort composed of Mexican–Mestizo BC patients only (Figure 9). Of them, only miR-1307-3p has been previously associated with chemoresistance [51] and resistance to cisplatin [52]. Other reports highlighted a downregulated activity of miR-940 in a Chinese cohort [53,54]. Such controversial studies highlight the importance of the participation of miRNAs as possible biomarkers in BC.

In summary, these results suggest the participation of a group of 61 MMRs that could be of high importance for the maintenance of tumor phenotype, as they are present in all the stages of the disease, and regulate a set of proved TSGs. Nevertheless, in vitro and in vivo experiments are required to prove their participation in BC carcinogenesis and as putative biomarkers.

## Figures and Tables

**Figure 1 cells-09-01610-f001:**
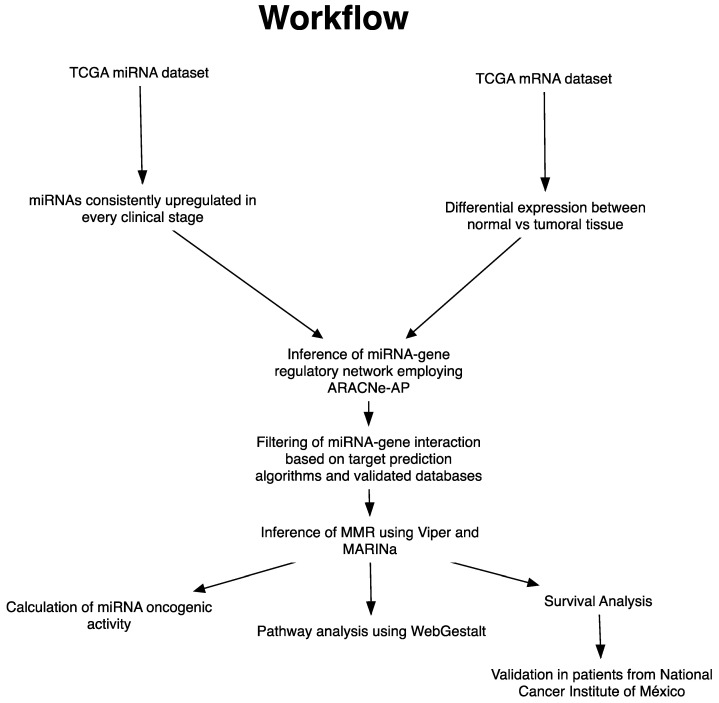
General workflow.

**Figure 2 cells-09-01610-f002:**
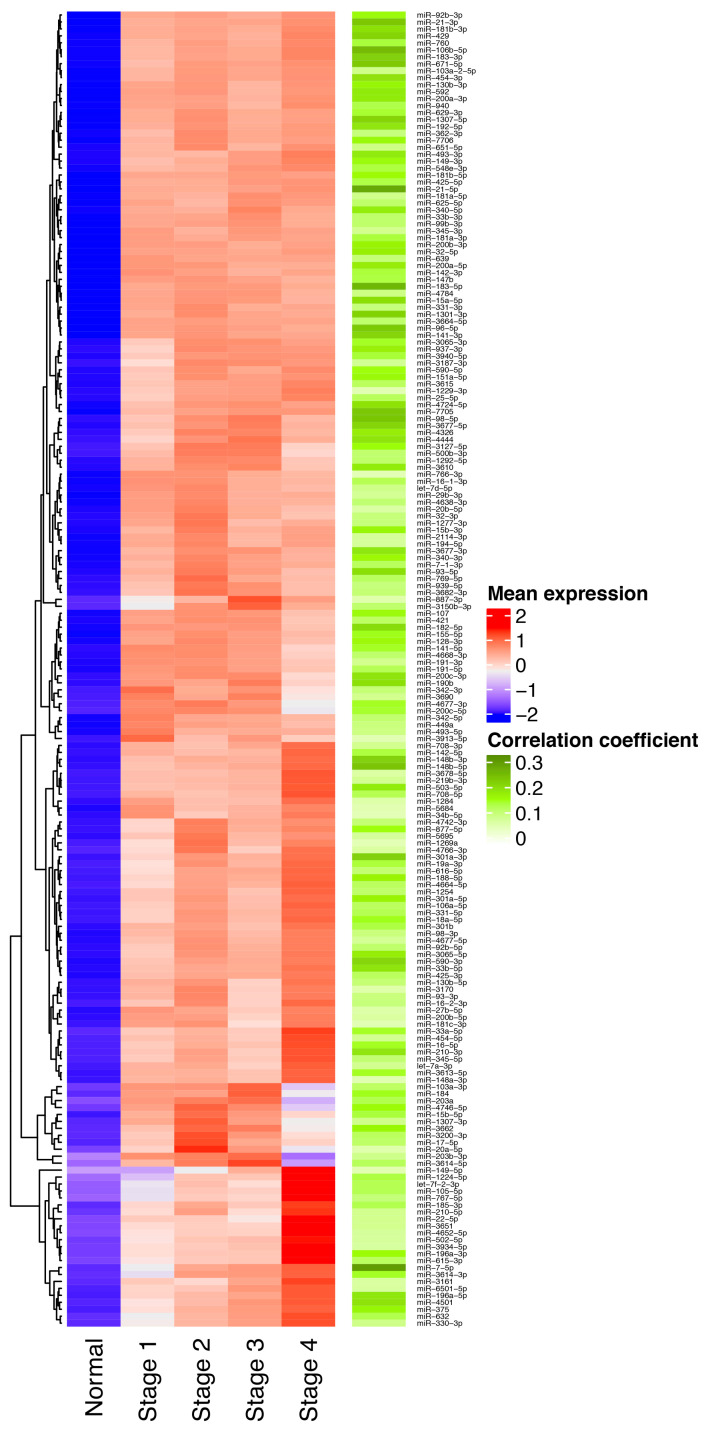
Overexpressed miRNAs across all clinical stages. The left heatmap shows the expression of 217 miRNAs over-expressed in breast cancer (BC) tissues (depicted in red) and downregulated in normal tissues (blue). As shown the majority of miRNAs had a positive correlation (green) between expression level and clinical stage.

**Figure 3 cells-09-01610-f003:**
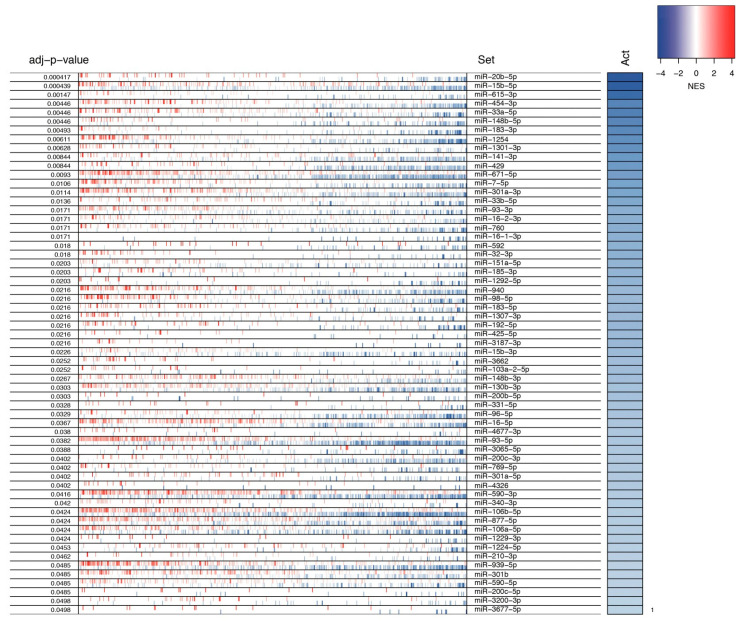
miRNA master regulators in breast cancer. The heatmap represents the log2 fold change expression of each miRNA target, where red shows an overexpression and blue a downregulation in BC tissues. The right bar shows the global regulation activity of each miRNA related to its multiple targets, represented as the normalized enrichment score (NES), where blue represents a negative NES and red a positive. As shown, all of the miRNAs exhibit a negative NES.

**Figure 4 cells-09-01610-f004:**
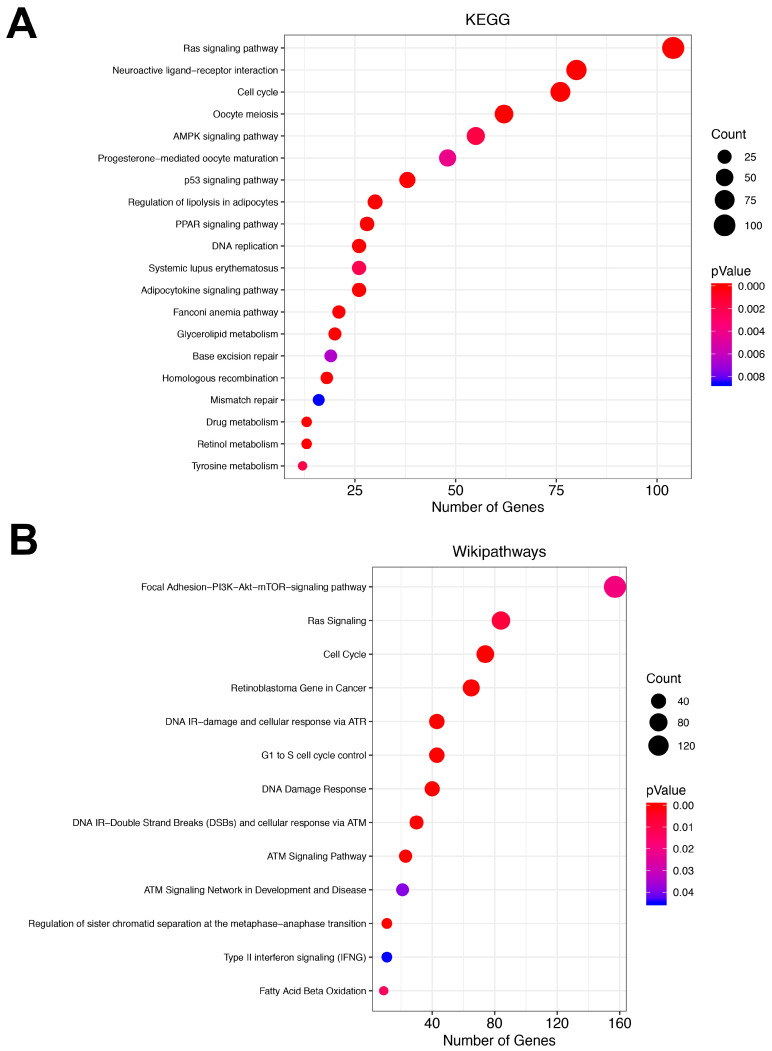
Pathway analysis in the WebGestalt platform. (**A**) Shows the enriched pathways in the KEGG database. The size of the dots depicts the number of genes in the pathway, while the dot color represents the significance of the analysis, where red shows a more significantly enriched pathway. (**B**) Shows the pathway enrichment in the Wikipathways database.

**Figure 5 cells-09-01610-f005:**
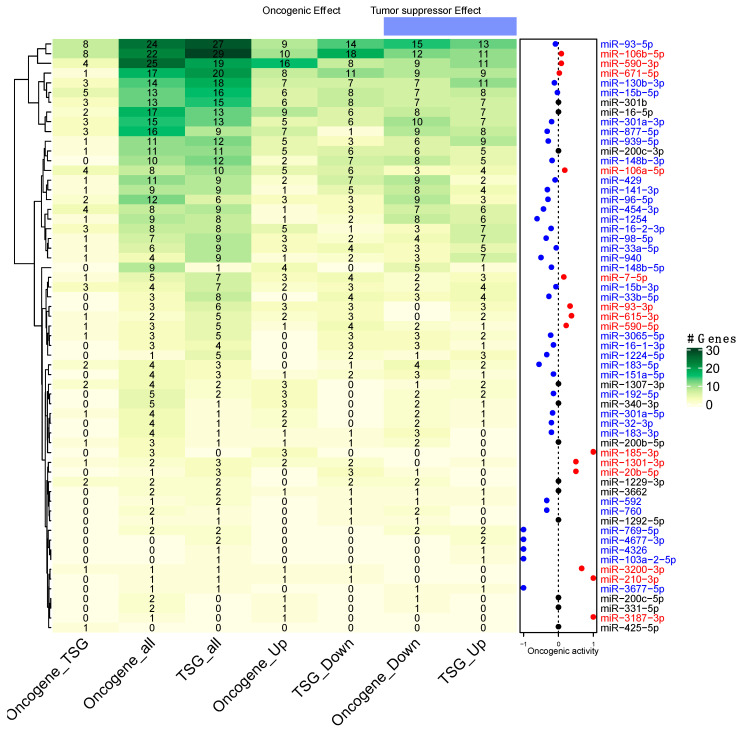
Oncogenic net activity of the 61 miRNAs. The heatmap shows the number of annotated oncogenes and tumor suppressors in the COSMIC database, where green represents a higher number of genes. The right dot plot represents each miRNA oncogenic activity, where blue shows a tumor suppressor activity, black a neutral activity, and red an oncogenic activity.

**Figure 6 cells-09-01610-f006:**
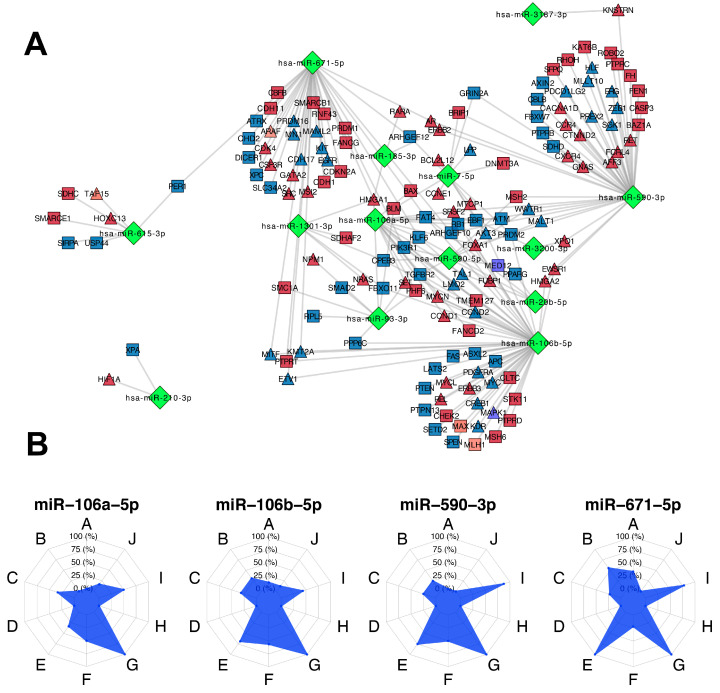
The 14 oncogenic miRNA master regulators (OncoMMRs) with a positive contribution to the tumoral phenotype and their driver targets. (**A**) The OncoMMR network where miRNAs are depicted as green diamonds, oncogenes as triangles and tumor suppressor genes as rectangles; blue and red represent downregulated and overexpressed genes, respectively. (**B**) Radar plots show the enrichment of genes as per the hallmark for the top four most highly connected OncoMMRs, where A: Angiogenesis, B: Cell replicative immortality, C: Change of cellular energetics, D: Evading immune response, E: Evading programmed cell death, F: Genome instability, G: Invasion and metastasis, H: Proliferative signaling, I: Suppression of growth, J: Tumor promoting inflammation.

**Figure 7 cells-09-01610-f007:**
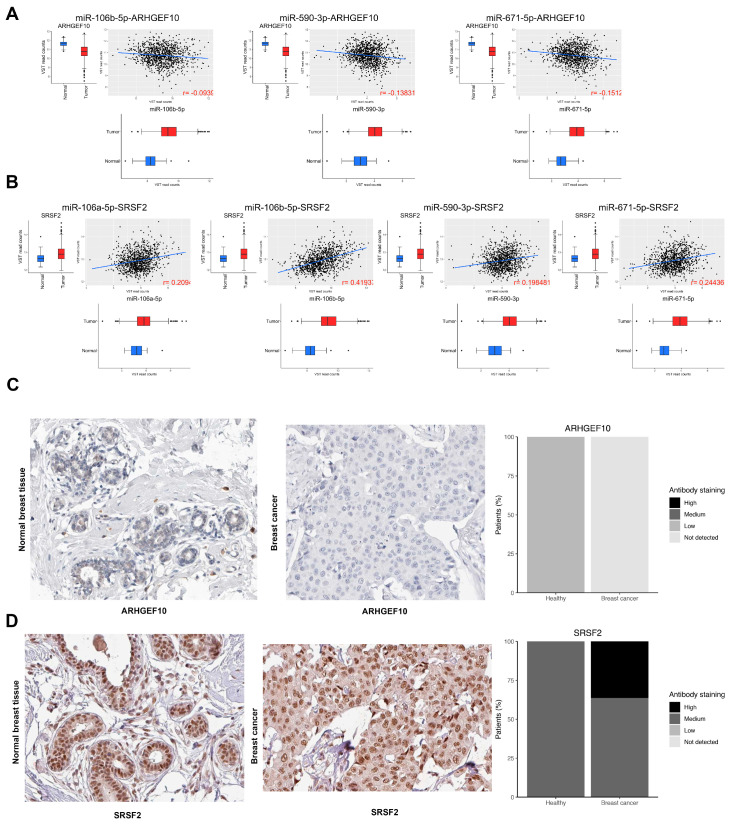
Correlation between the top OncoMMRs and the expression of the driver genes ARHGEF10 and SRSF2. (**A**) The correlation between the OncoMMRs and ARHGEF101. (**B**) The correlation between the OncoMMRs and SRSF2. Red boxplots represent the expression of the molecules in tumor tissues and blue in normal tissues. (**C**,**D**) protein expression of ARHGEF10 and SRSF2 from breast tissues of the human protein atlas.

**Figure 8 cells-09-01610-f008:**
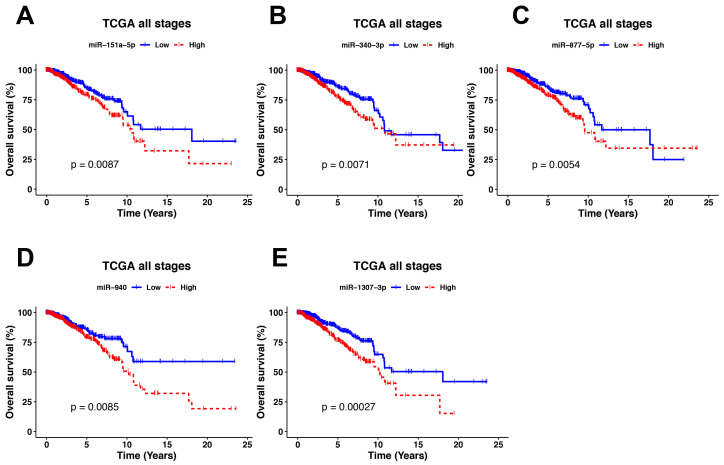
Kaplan–Meier overall survival analysis based on five clinically significant MMRs. (**A**) Shows the differences in the survival of patients with a high and low expression of miR-151a-5p. Blue lines represent patients with a low miRNA expression and red patients with a high expression. (**B**) Survival for miR-340-3p. (**C**) Survival for miR-877-5p. (**D**) Survival for miR-940 and (**E**) Survival for miR-1307-3p.

**Figure 9 cells-09-01610-f009:**
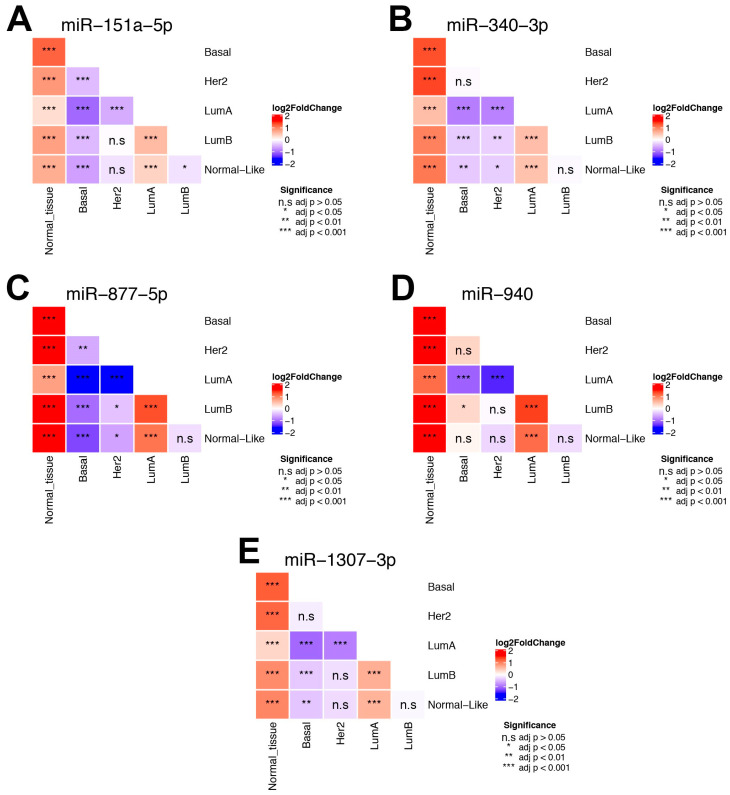
Differential expression of the five MMRs between PAM50 intrinsic subtypes. (**A**) Shows the differential expression of miR-151a-5p. The heatmap represents the log2FoldChange between each subtype comparison, where red shows a positive foldchange and blue a negative one. Significance is shown inside each cell, where * = adj *p* <0.05, ** = adj *p* < 0.01, *** = adj *p* < 0.001 and n.s = adj *p* > 0.05. (**B**) Represents miR-340, (**C**) Depicts miR-877 5p, (**D**) Shows miR-940 and (**E**) Represents the differential expression between subtypes of miR-1307-3p.

**Figure 10 cells-09-01610-f010:**
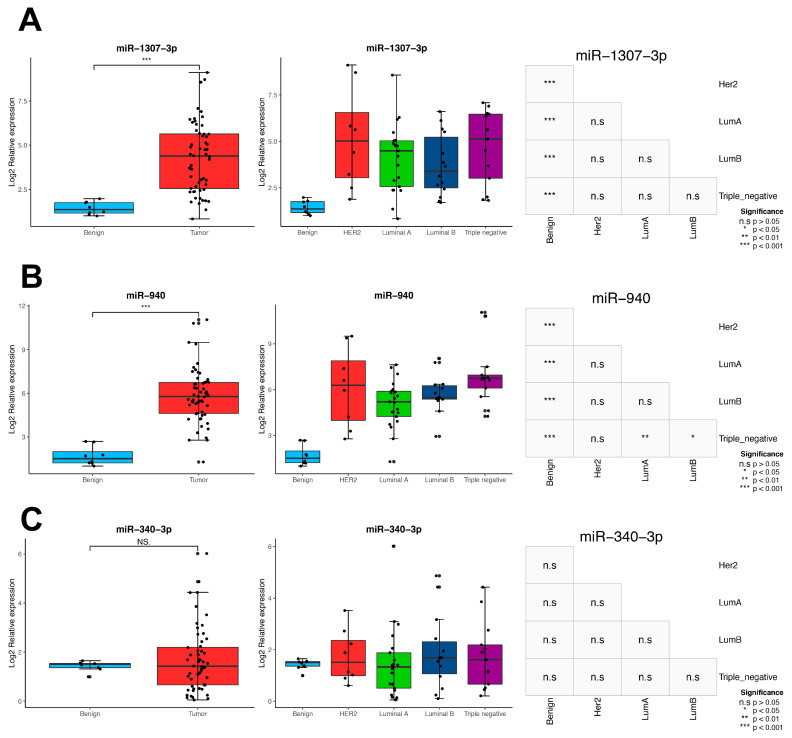
Validation in 56 locally advanced breast cancer samples using real-time qPCR. (**A**) Expression of miR-1307-3p in normal vs tumoral tissues and between the tumor subtypes. The boxplots represent the log2 relative expression and the heatmap shows the significance of the comparisons between each subtype, where the significance is annotated as * = *p* < 0.05, ** = *p* < 0.01, *** = *p* < 0.001 and n.s = *p* > 0.05. (**B**) Expression of miR-940 and (**C**) Expression of miR-340-3p.

**Table 1 cells-09-01610-t001:** Top degree miRNAs and genes in the network.

miRNAs.		Genes		
Name	Degree	Name	Degree	Gene log2FoldChange
hsa-miR-106b-5p	1179	RUNX1T1	24	−1.715126046
hsa-miR-590-3p	996	BNC2	20	−0.440592219
hsa-miR-93-5p	970	TNS1	20	−3.104826764
hsa-miR-671-5p	921	DLC1	18	−1.789077079
hsa-miR-939-5p	711	FOXP2	18	−1.963889919
hsa-miR-877-5p	670	IGF1	17	−2.494558934
hsa-miR-130b-3p	585	ZEB2	17	−1.552276689
hsa-miR-16-5p	547	ANK2	16	−2.347077977
hsa-miR-301a-3p	546	BACH2	16	−1.35469313
hsa-miR-15b-5p	529	TCF4	16	−0.872983915
hsa-miR-940	483	ZCCHC24	16	−1.552147667
hsa-miR-301b	469	ZFHX4	16	−1.12229256
hsa-miR-106a-5p	451	NOVA1	15	−0.712825148
hsa-miR-1254	398	TSHZ3	15	−0.478344046
hsa-miR-93-3p	373	BHLHE41	14	−1.138290499
hsa-miR-141-3p	355	CCND2	14	−0.965637676
hsa-miR-96-5p	338	CREBRF	14	−0.973637932
hsa-miR-454-3p	329	PRICKLE2	14	−1.12294618
hsa-miR-200c-3p	329	QKI	14	−1.25158266
hsa-miR-429	327	RACGAP1	14	1.909634723

**Table 2 cells-09-01610-t002:** OncoMMRs and their oncogenic activity.

OncoMMRs		
Name	Degree	Oncogenic Activity
miR-106b-5p	51	0.08
miR-590-3p	44	0.08
miR-671-5p	37	0.02
miR-106a-5p	18	0.18
miR-7-5p	12	0.15
miR-93-3p	9	0.33
miR-590-5p	8	0.22
miR-615-3p	7	0.37
miR-1301-3p	5	0.5
miR-20b-5p	4	0.5
miR-185-3p	3	1
miR-3200-3p	2	0.66
miR-210-3p	2	1
miR-3187-3p	1	1

**Table 3 cells-09-01610-t003:** Cox regression analysis of the overall survival of the breast cancer patients.

		Univariate Analysis		Multivariate Analysis	
	Overall Survival	HR (95% CI)	*p*-Value	HR (95% CI)	*p*-Value
miR-151a-5p		0.64 (0.46–0.9)	0.0093	0.76 (0.50–1.16)	0.2172
miR-940	High vs. low expression	0.63 (0.44–0.89)	0.0091	0.77 ( 0.53–1.14)	0.2026
miR-1307-3p		0.53 (0.38–0.75)	0.00033	0.63 (0.42–0.97)	0.0357
miR-340-3p		0.63 (0.45–0.89)	0.0076	0.91 (0.61–1.34)	0.6429
miR-877-5p		0.62 (0.44–0.87)	0.0059	0.98 (0.66–1.45)	0.9272
Age at diagnosis	<58 vs. ≥58	1.9 (1.3–2.6)	0.00038	2.12 (1.49–3.01)	2.78 × 10^−5^
ER	Negative vs. positive	0.57 (0.4–0.81)	0.0017	0.68 (0.39–1.19)	0.1802
PR		0.65 (0.47–0.91)	0.013	0.88 (0.51–1.50)	0.6473
Clinical stage		2.3 (1.8–2.8)	2.6 × 10^−12^	2.48 (1.96–3.14)	3.51 × 10^−14^

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
