# Peer review of "Identification of miRNA Master Regulators in Breast Cancer"

_cells, 2020, doi:10.3390/cells9071610_

Round 1
Reviewer 1 Report
The authors have submitted a revised version of their manuscript answering to my point of critics.
The manuscript has significantly been improved. However, I still have some remarks.
The multivariate cox regression analysis is helpful and partially answers the subgroup survival analysis, I have requested. However, I still do not understand why this subgroup-survival analysis has not been done. From the data presented, I see no evidence that the simple subgroups ER +/-, Her2-high and TNBC might not be different in this respect. In contrast, such data could further strengthen the point the authors want to make; namely that they have indeed identified “global“ regulators.
In the multivariate cox regression, why are there no data for ER and PR shown as in the univariate analysis?
Fig. 9 shows expression differences between the classical subgroups in the local samples, however the number of cases is very low (similar as in the proteinatlas-cases). The same analysis should be done with the TCGA dataset. The boxplots, however, suggest that there is no statistical difference between the subgroups but only statistical comparisons to the benign tissue are shown. Have the subgroups been tested for significance?
In Fig. 7C and D, some cases from proteinatlas.org are shown. This sample size is much too low to draw conclusions. It could be improved to do this analysis with the samples that have been used for the correlation analysis in Fig 7 A and B although this number is also very low (20).
Fig. 2 is scrambled in my copy of the manuscript. Please correct this.
Author Response
The manuscript has significantly been improved. However, I still have some remarks.
The multivariate cox regression analysis is helpful and partially answers the subgroup survival analysis, I have requested. However, I still do not understand why this subgroup-survival analysis has not been done. From the data presented, I see no evidence that the simple subgroups ER +/-, Her2-high and TNBC might not be different in this respect. In contrast, such data could further strengthen the point the authors want to make; namely that they have indeed identified “global“ regulators.
In the multivariate cox regression, why are there no data for ER and PR shown as in the univariate analysis?
RESPONSE: Thank you for your comment. We have addressed this issue by analyzing both the expression and survival of the 5 MMRs in each of the PAM50 intrinsic subtypes, this information was added in lines 366-371 and in the subsection 3.7 of the manuscript. We also have corrected the multivariate cox regression in table 3 with the corresponding values for PR and ER.
Fig. 9 shows expression differences between the classical subgroups in the local samples, however the number of cases is very low (similar as in the proteinatlas-cases). The same analysis should be done with the TCGA dataset. The boxplots, however, suggest that there is no statistical difference between the subgroups but only statistical comparisons to the benign tissue are shown. Have the subgroups been tested for significance?
RESPONSE: We have added the analysis of the TCGA dataset in section 3.7 of the manuscript and in the figure 9. Regarding the analysis in the local samples, we have increased the cohort to 56 samples which are shown in figure 10. In the previous version of the manuscript we stated in the figure legend that for clarity, only significant differences are annotated. In the revised version we updated the figure 10, showing all the p-values of each comparison as a heatmap, in addition to the boxplots.
In Fig. 7C and D, some cases from proteinatlas.org are shown. This sample size is much too low to draw conclusions. It could be improved to do this analysis with the samples that have been used for the correlation analysis in Fig 7 A and B although this number is also very low (20).
RESPONSE:. Regarding protein expression in tumor tissues, we understand that it is a limited number of samples but it is only one result in the whole manuscript. This result correctly illustrates the negative relationship between both proteins and the miRNAs analyzed. Furthermore, by increasing the number of samples from 20 to 56 the results were adequately confirmed.
Fig. 2 is scrambled in my copy of the manuscript. Please correct this.
RESPONSE: We have corrected the figure.
Reviewer 2 Report
The authors have began to address some of the concerns laid out previously, however these are only the minor suggestions. All of the major suggestions have been written off in the discussion. Whilst this softens the conclusions and helps with one of the suggestions, the remaining major concerns have not been addressed.
Although I understand why you would look for miRNAs that a pan-breast cancer, biologically this doesn't make sense. Breast cancer subtypes arise, mostly likely, through different cell types and different mechanisms, therefor the likelihood of a master miRNA regulator is very low and of less interest than subtype specific miRNAs.
In addition, citing a reference, Oliveira, doesn't justify not making the criteria more stringent regarding the miRNA databases.
There is a list of references to include, please include these. Accurate reference is essential in scientific work.
Author Response
The authors have began to address some of the concerns laid out previously, however these are only the minor suggestions. All of the major suggestions have been written off in the discussion. Whilst this softens the conclusions and helps with one of the suggestions, the remaining major concerns have not been addressed.
Although I understand why you would look for miRNAs that a pan-breast cancer, biologically this doesn't make sense. Breast cancer subtypes arise, mostly likely, through different cell types and different mechanisms, therefor the likelihood of a master miRNA regulator is very low and of less interest than subtype specific miRNAs.
RESPONSE: Thank you for your comment. Although the intrinsic subtypes show different phenotype characteristics such as a higher proliferation rate that impact in the clinical outcome there is evidence that shows the existence of similarities between them, such as the deregulation of common pathways and genes; for example in Bower et al., all of the subtypes displayed an aberrant cell cycle [1], while in Adams et al., they observed an increased activation of the PI3K enzyme in all of the subtypes [2] , among others.
Another fact that lead us to investigate the possible existence and impact of miRNAs master regulators expressed in all subtypes, was in part by the work of Perou et al [3], where the authors showed 496 genes that defined the intrinsic subtypes, although this list was further refined to 50 genes by Bernard et al, [4]. Considering this, and the fact that currently there are approximately 22000 annotated protein coding genes and that specifically in the TCGA breast cancer data the number of significant differentially expressed protein-coding genes in all tumors when compared with normal samples is around 14449 (Supplementary file 8) we reasoned that there could be a common “core” of genes and miRNAs shared by every subtype that could be responsible of maintaining the core mechanisms that a tumor needs in every step of the tumorigenesis and that differentiates them from normal healthy tissue; such as the hallmarks of resisting cell death, sustaining proliferative signaling, enabling replicative immortality and evading growth suppressors.
Another advantage of analyzing miRNAs expressed in all of the subtypes underlies in the possibility of using them as possible biomarkers in a wide cohort of patients. We added this information in the discussion in lines: 487-492 and added the section 3.7 in the manuscript where we analyze the expression of the miRNAs in the intrinsic subtypes of the TCGA samples. We highly appreciate your valuable comment as it improved our manuscript.
In addition, citing a reference, Oliveira, doesn't justify not making the criteria more stringent regarding the miRNA databases.
Response: In addition to our previous response, in Sethupathy et al., [5] it is stated that “the union of all programs achieves the highest sensitivity and the lowest specificity, likewise, the intersection of all programs achieves the highest specificity and the lowest sensitivity”, and suggest the use of the union of all prediction programs”. Moreover, in Ritchie et al., [6], the authors suggest that “the routine identification of an overlap between miRNA target prediction algorithms should be discouraged owing to a lack of utility and rationale” and propose that predictions should be filtered by the co-expression of the miRNA and the target, which is what we used in this study by considering only those predictions that appeared in our inferred mutual-information miRNA-mRNA network.
There is a list of references to include, please include these. Accurate reference is essential in scientific work.
RESPONSE: We have added the references in lines 46-48, 58, 468 and 526 .
REFERENCES
- Bower, J.J.; Vance, L.D.; Psioda, M.; Smith-Roe, S.L.; Simpson, D.A.; Ibrahim, J.G.; Hoadley, K.A.; Perou, C.M.; Kaufmann, W.K. Patterns of cell cycle checkpoint deregulation associated with intrinsic molecular subtypes of human breast cancer cells. npj Breast Cancer 2017, 3, 1–12. https://doi.org/10.1038/s41523-017-0009-7
- Adams, J.R.; Schachter, N.F.; Liu, J.C.; Zacksenhaus, E.; Egan, S.E. Elevated PI3K signaling drives multiple breast cancer subtypes. Oncotarget 2011, 2, 435–447. https://doi.org/10.18632/oncotarget.285
- Perou, C.M.; Sørile, T.; Eisen, M.B.; Van De Rijn, M.; Jeffrey, S.S.; Ress, C.A.; Pollack, J.R.; Ross, D.T.; Johnsen, H.; Akslen, L.A.; et al. Molecular portraits of human breast tumours. Nature 2000, 406, 747–752. https://doi.org/10.1038/35021093
- Bernard, P.S.; Parker, J.S.; Mullins, M.; Cheung, M.C.U.; Leung, S.; Voduc, D.; Vickery, T.; Davies, S.; Fauron, C.; He, X.; et al. Supervised risk predictor of breast cancer based on intrinsic subtypes. J. Clin. Oncol. 2009, 27, 1160–1167. https://doi.org/10.1200/JCO.2008.18.1370
- Sethupathy, P.; Megraw, M.; Hatzigeorgiou, A.G. A guide through present computational approaches for the identification of mammalian microRNA targets. Nat. Methods 2006, 3, 881–886. https://doi.org/10.1038/nmeth954
- Ritchie, W.; Flamant, S.; Rasko, J.E.J. Predicting microRNA targets and functions: Traps for the unwary. Nat. Methods 2009, 6, 397–398. https://doi.org/10.1038/nmeth0609-397
Round 2
Reviewer 1 Report
The authors have submitted a revised Version of their manuscript and responded to all my critics.
I think the paper has again been imrpoved significantly and from my point of view can now be published.
Reviewer 2 Report
Although I don't agree with the approach the authors have taken regarding the subtypes of breast cancer and the use of only a single database whilst citing many. They have provided enough scientific rational to have a compelling argument that the manuscript be published in it's current form.
This manuscript is a resubmission of an earlier submission. The following is a list of the peer review reports and author responses from that submission.
Round 1
Reviewer 1 Report
In this paper the authors take the task of identifying master regulator miRNAs for breast cancer by bioinformatics means and proving these results by employing a small cohort of patients. As data source, public databases (mainly TCGA) were used. After filtering for differential expression, the oncogenic activity of miRNAs was estimated by correlation with upregulated oncogenes and downregulated tumor suppressor genes. About 200 miRNAs were significantly correlated with staging.
miRNA targets were taken from several databases containing both, verified and hypothetical targets. Altogether this resulted in 65 putative master regulators.
The paper is generally well written and the bioinformatics part seems to be well done. I am not a real expert for these methods, but to me it looks quite convincing.
However, I have a few points that need to be improved from my point of view.
Databases containing miRNA targets are full with unverified and hypothetical target genes. Could the authors comment on the usefulness of dealing with these hypothetical data?
Breast cancer is a very heterogeneous entity; for the initial data collection the authors did not care about this and I think this is OK for screening, however I think for the final expression and survival analysis a subgroup analysis would be recommended. When analyzing all cases together about 85% will be ER-positive and the data from these cases will dominate the statistical outcome. I strongly recommend that the authors analyze the impact of the putative master regulators for specific subgroups, such as triple negative, HER2-over-expressing and lobular carcinomas.
As a second cohort, 20 patient samples were analyzed for miRNA /RNA expression. However, immunohistochemistry data were taken from the proteinatlas.org website. I am not sure what this should demonstrate. This web site shows only a few cases. It would be much better using the 20 patient samples for staining and correlate these data with the miRNAs/mRNA results from exactly these patients.
Reviewer 2 Report
Martinez-Gutierrez and Pérez-Plasencia et al attempt to define a set of miRNAs that act as master regulators to control global transcriptional profiles in breast cancer independent of molecular subtypes. This concept is a great one, miRNAs probably do have this potential to have far reaching consequences on the transcriptome as a whole, however this is not a trivial hypothesis and a very challenging one to address probably. Unfortunately the authors have largely fallen short of doing this and the work submitted here would require major changes to be accurate and relevant to the breast cancer community.
Major:
- The overall grammar and spelling of the submitted work is very poor. There are numerous spelling mistakes throughout the manuscript, poor choice or wording and incorrectly structured sentences. These need to dramatically change for a journal that publishes English language manuscripts.
- Example, sentence in abstract is the same as a sentence in the introduction:
- Line 19, “In the present study our major aim was the identification of miRNAs acting as master regulators that were consistently up-regulated along the clinical stages of breast cancer regardless molecular subtypes. To achieve this, we downloaded from TCGA miRNA and mRNA sequencing data corresponding to 1091 breast cancer patients and 110 normal adjacent tissues. Clinical stage information was employed to categorize each patient and expression data was compared to normal tissues.”
- Line 57, “In the present work, our major aim was the identification of miRNAs acting as master regulators that were consistently up-regulated along the clinical stages of BC regardless molecular subtypes. To achieve this, we downloaded from TCGA miRNA and mRNA sequencing data corresponding to 1091 breast cancer patients and 110 normal adjacent tissues. Clinical stage information was employed to categorize each patient and expression data was compared to normal tissues.”
- The conclusions are not well founded. The bioinformatic work they have done doesn’t confirm anything, merely finds associations in expression patterns between molecules. For example:
- Line 270, “ARHGEF10 and the oncogene SRSF2 were commonly regulated by three (miR…”
- These three miRNAs have not been proven to regulate either one of these genes, neither have most of the miRNA-mRNA pairs, at most they are associations and predictions, as very few miRNA-mRNA pairs have been experimentally validated. The worded must be adjusted to convey this message accurately to the reader.
- Line 361-364, the IHC seen in figure 7 merely shows that these two genes can be over-expressed in cancer. As far as I can tell this does not qualify as validation their either ARHGEF10 or SRSF1 are controlled by miRNAs in breast cancer.
- The study design is majorly flawed. Why would you look for miRNAs that are master regulators of transcriptomes independent of breast cancer subtype when we know that the subtypes are what define the tumour transcriptomes? Seems like the authors have taken an easier way out by doing this and if the study was to be conducted on a subtype basis would find more meaningful results.
- The criteria for picking miRNA-Gene interactions is far too relaxed. Given that the authors employed the use of 11 different databases but their criteria for picking a miRNA-Gene interaction was that it had to be present in at least 1 database, would suggest that far too many false-positives are included in the data. These criteria should be boosted to be half of the prediction databases, and priority given to those that also appear in the experimentally validated databases.
- This work does not cite important references. Major advances have been made in the field of miRNAs in breast cancer research and the paper by Dvinge, listed below, especially needs to be discussed as there is much overlap in the work proposed here to that landmark study. In addition to this, each of the miRNAs that are focused upon in the prognostic section should be referenced, the authors can’t look into every single miRNA, but they do need to better acknowledge that many of their miRNAs of interest have already been well studied in cancer and particularly breast cancer.
- V. Iorio, M. Ferracin, C. G. Liu, A. Veronese, R. Spizzo, S. Sabbioni, E. Magri, M. Pedriali, M. Fabbri, M. Campiglio, S. Menard, J. P. Palazzo, A. Rosenberg, P. Musiani, S. Volinia, I. Nenci, G. A. Calin, P. Querzoli, M. Negrini, C. M. Croce, MicroRNA gene expression deregulation in human breast cancer. Cancer Res 65, 7065-7070 (2005).
- Volinia, G. A. Calin, C. G. Liu, S. Ambs, A. Cimmino, F. Petrocca, R. Visone, M. Iorio, C. Roldo, M. Ferracin, R. L. Prueitt, N. Yanaihara, G. Lanza, A. Scarpa, A. Vecchione, M. Negrini, C. C. Harris, C. M. Croce, A microRNA expression signature of human solid tumors defines cancer gene targets. Proceedings of the National Academy of Sciences of the United States of America 103, 2257-2261 (2006).
- Volinia, M. Galasso, M. E. Sana, T. F. Wise, J. Palatini, K. Huebner, C. M. Croce, Breast cancer signatures for invasiveness and prognosis defined by deep sequencing of microRNA. Proceedings of the National Academy of Sciences of the United States of America 109, 3024-3029 (2012).
- Cascione, P. Gasparini, F. Lovat, S. Carasi, A. Pulvirenti, A. Ferro, H. Alder, G. He, A. Vecchione, C. M. Croce, C. L. Shapiro, K. Huebner, Integrated microRNA and mRNA signatures associated with survival in triple negative breast cancer. PloS one 8, e55910 (2013).
- Gasparini, L. Cascione, M. Fassan, F. Lovat, G. Guler, S. Balci, C. Irkkan, C. Morrison, C. M. Croce, C. L. Shapiro, K. Huebner, microRNA expression profiling identifies a four-microRNA signature as a novel diagnostic and prognostic biomarker in triple negative breast cancers. Oncotarget 5, 1174-1184 (2014).
- Dvinge, A. Git, S. Graf, M. Salmon-Divon, C. Curtis, A. Sottoriva, Y. Zhao, M. Hirst, J. Armisen, E. A. Miska, S. F. Chin, E. Provenzano, G. Turashvili, A. Green, I. Ellis, S. Aparicio, C. Caldas, The shaping and functional consequences of the microRNA landscape in breast cancer. Nature 497, 378-382 (2013).
- Volinia, C. M. Croce, Prognostic microRNA/mRNA signature from the integrated analysis of patients with invasive breast cancer. Proceedings of the National Academy of Sciences of the United States of America 110, 7413-7417 (2013).
- All of the figure legends are a severe lack of detail, I often had to read them many times or read the text to figure out what they were trying to say. These need a serious overhaul to be more informative to the reader.
- The pathway analysis if flawed here as there are over 6000 genes going into the analysis, so of course many interesting pathway will be found. All of this analysis needs to be redone when the authors define a smaller more robust set of miRNA-Gene interactions as suggested above.
- Many of the images are to pixelated for me to read so I can’t comment of many figures.
- The oncogenic activity seen in figure 5, can a significance be applied to this as most of the miRNAs has a very small change which is probably not at all meaningful.
- It also seems very strange that most of these miRNAs are tumour suppressor genes, given that that are all over-expressed in cancer and their expression increases with stage, this is a common feature of oncogenes (Figure 2). If the analysis is redone with subtypes this should be re-explored.
- Example, sentence in abstract is the same as a sentence in the introduction:
Minor:
- The original TCGA paper needs to be cited in the methods section
- Cancer Genome Atlas, Comprehensive molecular portraits of human breast tumours. Nature 490, 61-70 (2012).
- Line 79, fold change of 0.5, what do the authors mean by this? Fold change of log2 1.5 or 0.5? +/-0.5?
- Line 80, TNM not defined
- Line 142, new paragraph missing
- Line 143, where was the clinical data obtained?
- Line 145, reference missing for R package
- Line 161, full qPCR method missing, write out in full please
- Line 199, 27,940 De genes seems like a lot, there are almost 60,000 genes in that table, are there a whole heap of predicted genes that should be removed?
- Figure 3. No scale bars, what kind of expression are we look at in the figure, RPKK, mean-centred, Z-scores? Is the expression average between normal adjacent vs tumour?
- Line 275, what trend are the authors referring to here? The fact that ARHGEF10 and SRSF2 are over-expressed in cancer?
- Line 290, there are not enough clinical features included int eh univariate and multivariate analysis to discern the impact these miRNAs have on prognostication. Please at least include tumour size, IHC status of ER, PR, HER2 and lymph node status.
- Figure 8, please include patient numbers, hazards ratios and confidence intervals.
- Citations missing on line 326 and 346.